# Intermolecular Mechanism and Dynamic Investigation of Avian Influenza H7N9 Virus’ Susceptibility to E119V-Substituted Peramivir–Neuraminidase Complex

**DOI:** 10.3390/molecules27051640

**Published:** 2022-03-02

**Authors:** Sphamandla E. Mtambo, Samuel C. Ugbaja, Aganze G. Mushebenge, Bahijjahtu H. Abubakar, Mthobisi L. Ntuli, Hezekiel M. Kumalo

**Affiliations:** 1Drug Research and Innovation Unit, Discipline of Medical Biochemistry, School of Laboratory Medicine and Medical Science, University of KwaZulu-Natal, Durban 4000, South Africa; sphamtambo@gmail.com (S.E.M.); aganzedar@gmail.com (A.G.M.); 2Renewable Energy Programme, Federal Ministry of Environment, 444 Aguiyi Ironsi Way, Maitama, Abuja 904101, Nigeria; bahijjah@yahoo.com; 3Department of Mathematics, Faculty of Applied Science, Durban University of Technology, Durban 4001, South Africa; mthobisin2@dut.ac.za

**Keywords:** peramivir, neuraminidase, H7N9, influenza, virus, hemagglutinin, mutation, E119V

## Abstract

The H7N9 virus attaches itself to the human cell receptor protein containing the polysaccharide that terminates with sialic acid. The mutation of neuraminidase at residue E119 has been explored experimentally. However, there is no adequate information on the substitution with E119V in peramivir at the intermolecular level. Therefore, a good knowledge of the interatomic interactions is a prerequisite in understanding its transmission mode and subsequent effective inhibitions of the sialic acid receptor cleavage by neuraminidase. Herein, we investigated the mechanism and dynamism on the susceptibility of the E119V mutation on the peramivir–neuraminidase complex relative to the wildtype complex at the intermolecular level. This study aims to investigate the impact of the 119V substitution on the neuraminidase–peramivir complex and unveil the residues responsible for the complex conformations. We employed molecular dynamic (MD) simulations and extensive post-MD analyses in the study. These extensive computational investigations were carried out on the wildtype and the E119V mutant complex of the protein for holistic insights in unveiling the effects of this mutation on the binding affinity and the conformational terrain of peramivir–neuraminidase E119V mutation. The calculated total binding energy (ΔG_bind_) for the peramivir wildtype is −49.09 ± 0.13 kcal/mol, while the E119V mutant is −58.55 ± 0.15 kcal/mol. The increase in binding energy (9.46 kcal/mol) is consistent with other post-MD analyses results, confirming that E119V substitution confers a higher degree of stability on the protein complex. This study promises to proffer contributory insight and additional knowledge that would enhance future drug designs and help in the fight targeted at controlling the avian influenza H7N9 virus. Therefore, we suggest that experimentalists collaborate with computational chemists for all investigations of this topic, as we have done in our previous studies.

## 1. Introduction

The influenza virus has been considered one of the most common respiratory diseases. The avian influenza virus has three popular types: A, B and C viruses. The presence of hemagglutinin (HA) and neuraminidase (NA) proteins in A and B types makes them treatment targets for drug inhibition [1,2]. These viruses spread from human to human through droplets and aerosol contents of the coughs and sneezes of an infected human. They enter the body’s cells through the upper respiratory tract in a process called endocytosis [3]. The common symptoms of influenza virus infection include headache, sore throat, fever, cough, muscle aches, catarrh and general weakness. The infection can lead to diseases in other parts of the body, such as the heart, lungs, and central nervous system [4]. Influenza A and B viruses are made up of single genomic strands of ribonucleic acid (RNA), which is needed for the viral polymerase to replicate within the cell of the infected person. In addition, they comprise viral glycoproteins, such as hemagglutinin and neuraminidase, which regulate the entrance and exit of the virus to and from the cell. Other components include matrix protein M1, membrane protein M2, nuclear export protein (NEP), non-structural protein (NS1) and viral nucleoprotein (NP) [5,6].

According to the World Health Organization, the first human case of avian influenza A H7N9 was reported in China on the 31st of March, 2013. Good knowledge of its mode of transmission is a prerequisite for effective management of the situation. The H7N9 virus attaches itself to the human cell receptor protein containing the polysaccharide which terminates with sialic acid. The upper respiratory tract predominantly consists of α-2,6-link receptors where the H7N9 virus attaches [7,8]. The HA has trimeric subunits; each has one domain that goes through the viral envelop domain while the other attaches through the sialic acid receptor on the human cell. For the H7N9 to effectively enter the host cell, the hemagglutinin HA must be cleaved; this cleaving is facilitated by an enzyme released from the epithelial lining of the human respiratory tract. This cleaving of the HA produces a hydrophobically hidden fusion peptide in the HA trimeric complex [6]. The host cell takes up the virus through endocytosis, and the endocytic vesicles combine with the lysosomes and result in its internal acidification (lowers its pH). This lowering of pH causes backwards shifting in the receptor binding conformation, thereby activating the forward shifting of the fusion peptide to enter the vesicular membrane. This fusion process discharges the contents of the virus into the cytoplasm, making the H7N9 virus fit for the replication cycle [9].

Neuraminidase’s significant role in releasing the H7N9 virus that has already entered the human host cell through endocytosis makes them the most common human avian H7N9 virus therapeutic target known to date. Neuraminidase splits sialic acid, thereby allowing the trapped virus to exit the cell. The neuraminidase inhibition target aims to halt the splitting of sialic acid and subsequent prevention of the exit of the virus from the human cell. This prevention of the H7N9 virus from exiting the cell deprives it of accessing the required resources for replication, thereby directly stopping its activities [10,11]. Neuraminidase inhibitor (NAI) binds to NA, preventing the virus from leaving the cell [12]. The NAI can only inhibit influenza type A and B and not C because type C does not possess the neuraminidase protein [13]. Therefore, the World Health Organization approved NAIs as the most effective drugs for treating influenza viral infections since 2010. The approved neuraminidase drugs (Figure 1) include oseltamivir, zanamivir and laninamivir [14,15]. Donald et al. (2010) carried out combinatory research on peramivir and oseltamivir. The conducted experimental research studied the chemotherapeutic effects of the combined treatment on mice infected with the influenza virus. The treatment was carried out with peramivir and oseltamivir for three days. The result showed that such combination yielded an improved outcome, better than treatment with suboptimal doses of peramivir and oseltamivir, in the treatment of influenza virus in mice [16]

The later approval of an intravenous highly selective potent peramivir (Figure 1) by the Food and Drug Administration has positively contributed to the fight against the influenza H7N9 virus. Hseir et al. (2021) investigated the efficacy of peramivir over oseltamivir in an emergency unit. They found that patients administered with a single dose of peramivir exhibited a quicker and better recovery than those administered with oseltamivir. The structure of peramivir consists of carboxylic and guanidino moieties in addition to the lipophilic side chain of the cyclopentane backbone. These structural moieties enable peramivir to establish a stronger bond with the N9 of the neuraminidase enzyme with a low dissociation rate. Zhang et al. investigated the pharmacokinetic properties of peramivir in selected healthy Chinese volunteers. Three hundred to four hundred milligram doses of peramivir were intravenously administered to them. When blood samples were collected at intervals after administering the doses, it was discovered that peramivir exhibited linear pharmacokinetics with an observed increase in maximum concentration. In another study by Sato et al., aimed at determining and predicting the peramivir concentration–time curve of the H7N9 virus, a reduced susceptibility to neuraminidase inhibitors was observed. Ten milligram per kilogram of peramivir was intravenously administered to paediatric patients; peramivir resulted in reduced maximal concentration of about 0.1 percent within 24 h of the drug administration. The study suggested re-administering peramivir in cases of delayed improvement in patients with normal susceptibility to influenza A and B, and that better viral inhibition and lower frequency of adverse effects may be expected with divided administration [17,18,19,20,21,22,23]. This further informed our aim to investigate the interatomic and intermolecular interactions of peramivir to ascertain the reasons for this observation [24]. Generally, in both A and B influenza virus types, neuraminidase proteins have characteristic conserved active site residues. The conserved catalytic residues in direct contact with sialic acid are Arg118 (R118), Asp 151 (D151), Arg152 (R152), Arg224 (R224), Glu276 (E276), Arg292 (R292), Arg371 (R371) and Tyr406 (Y406). In addition, other enzymes are supporting binding domain framework residues, which include Glu119 (E119), Arg156 (R156), Trp178 (W178), Ser179 (S179), Asp198 (D198), Ile222 (I222), Glu227 (R227), His274 (H274), Glu227 (R227), Asn294 (N294) and Glu425 (R425) [25,26,27].

Transcription of DNA leads to messenger RNA, and subsequent translation of messenger RNA leads to protein formation. Therefore, mutation is an alteration in gene structure as a result of a variant form, which could be transmitted to the next generations [28]. This results from a change in either one of the base units of the DNA or when a more significant portion of the genes/chromosomes are rearranged or deleted [28]. The mutation of any central region of the conserved active domain of the amino acid affects the responsiveness of the human H7N9 virus to NAIs, thereby resulting in resistance to the treatment potencies of the NAIs. Previous studies have investigated the consequences of introducing or altering amino acid residues on the responsiveness and susceptibility of the human H7N9 virus on NAIs [29,30].

Jing et al. studied the avian influenza H7N9 virus’ replicability and susceptibility potentials to NAIs when some amino acid residues were substituted, such as Arg292 to Lys292 (R292K), and Glu 119 to Val119 (E119V). The result showed that the neuraminidase inhibitor-resistant mutation in H7N9 could only reduce the susceptibility of neuraminidase inhibitors while the replicability remained unaffected [31]. The single mutation, E119V, has clinically been reported in oseltamivir, an analogue of peramivir. However, the molecular dynamics and conformational alterations resulting from E119V substitution on peramivir is yet to be investigated. Therefore, it is imperative to further study the effects of this mutation on peramivir, an analogue of oseltamivir at the intermolecular level [14,32,33]. The focus of this present study is to computationally investigate the intermolecular mechanisms of E119V-substituted NA on peramivir. We employed conventional molecular dynamic simulations MD of 250 nanoseconds on the wildtype (WT) and the E119V mutation, and carried out subsequent post-MD analyses, such as root mean square deviation (RMSD), root mean square fluctuation (RMSF), hydrogen bond analysis, binding free energy calculations and the radius of gyration and principal component analysis (PCA) [34,35]. This study promises to offer contributory insight and additional knowledge to enhance future drug designs and help in the fight targeted at controlling the avian influenza H7N9 virus.

## 2. Materials and Methods

### 2.1. System Preparation

The X-ray crystallographic structure of neuraminidase–peramivir (wildtype) complex was retrieved from RCSB protein data bank with PDB ID 4MWV. The chimera software was applied in the mutation of arginine to valine (E119V), as well as in the ligand and protein preparations. The ligand was separated from the receptor and hydrogen and charges were added to it. The ligand was saved as mols. The other nonstandard residues and water were removed from the receptor, which was saved as PDB.

### 2.2. Molecular Dynamic Simulations

Molecular dynamic simulations provide a resemblance of possible real time situations, used in predicting the probability of ligand–protein or protein–protein interactions in a time period. It involves the computation of Newton’s laws of motion and quantum physics in predicting the way atoms behave in three-dimensional space [36]. Molecular dynamic simulations of 250 ns were performed on both the wildtype of the peramivir–neuraminidase complex and the E119V mutant. An Amber 18 graphic processing unit (GPU) particle mesh Ewald molecular dynamic (PMEMD) was employed in the MD simulations. The force-field-related parameters and protein description were handled with FF14SB [37,38]. LEAP module implementation of Amber18 was utilised in the addition of hydrogen atoms to the neuraminidase (protein) and subsequent counter-ions additions to neutralise the system [39]. The systems were restrained in TIP3P water box, having the protein atoms situated at eight angstrom (8 Å) distance away to the end of the water box [40]. The system employed the periodic boundary conditions and the handling of the long-range electrostatics, using PMEMD in Amber18 with twelve angstrom (12 Å) van der Walls cut off. The initial minimisation was conducted using a restrained potential of 500 kcal/mol/Å^2^ in 1000 steepest descent steps and 1000 conjugate gradient steps on the solute [41]. This was followed by 1000 steps unrestrained conjugate gradient minimisation for the entire system. A gradual heating (0 K to 300 K) used NVT canonical ensemble and a harmonic restraint of 5 kcal/mol/Å^2^ for the solute atoms with a one pico-second random collision frequency. An unrestrained equilibration of the system, using NPT ensemble at 1 bar and 300 K, was performed [42]. A molecular dynamics simulation production run of 250 ns was conducted using an isothermal isobaric (NPT) ensemble and a Berendsen barostat [43]. The coordinates were saved at intervals after each stage and the trajectories were analysed. The post-MD analyses were carried out with Amber18 implemented modules, PTRAJ and CPPTRAJ. The plots were conducted using origin software, while the visualisations were conducted with chimera molecular modelling [44]. We measured system stability through the root mean square deviation (RMSD) calculation.

## 3. Results and Discussion

The results of the post-MD analyses, such as root mean square deviation (RMSD), root mean square fluctuation (RMSF), radius of gyration (RoG), solvent accessible surface area (SASA), hydrogen bond analysis, binding free energy calculations and principal component analysis (PCA), which were calculated for holistic knowledge on the effects of mutations on binding and the conformational terrain of the complex (peramivir-neuraminidase, are briefly discussed below).

### 3.1. Root Mean Square Deviations (RMSDs)

The root mean square deviation trajectory of the protein backbone alpha carbon (Cα) was produced using the CPPTRAJ module [45]. The standard deviation of the interatomic distance between the Cα backbone atoms of two amino acids were determined based on Equation (1):(1)$$RMSD\left(v,w\right)=\sqrt{\frac{1}{n}\overset{n}{\underset{i=1}{\sum }}||v_{i}-w_{i}}|{|}^{2}\;$$
where *n* is the number of atoms, *v**_i_* is the coordinate vector for target atom *I*, and *w**_i_* is the coordinate vector for reference atom *i*. RMSD measures how the target coordinate set deviates from the reference coordinate sets. The root mean square deviation measures the similarities of the structures of biomolecular compounds and their dynamism. The root mean square deviation of backbone C-α atoms was evaluated (Figure 2), and the conformation’s stability was analysed for peramivir wildtype and E119V mutant systems. We also observed the alignment of all the protein frames using the wildtype complex as the reference frame backbone. This provides insight and additional information into the root mean square deviation evolutionary trend of the protein’s structural conformation during the simulation run. Figure 2 depicts the evaluated root mean square deviation values calculated for the two systems. We observed relative stability in both complexes, from 10 ns to 115 ns, with the mutant complex exhibiting higher stability. However, the mutant complex showed unusual higher values of 2.5 Å to 2.7 Å (at 115–149 ns), which could be due to the system exhibiting lower binding affinity at this time frame. Subsequently, the two complexes maintained good stability, with the wildtype complex displaying higher values from 150 ns to the end of the simulation time frame.

These results suggested an overall stable molecular dynamics trajectory, occurring in an acceptable range during the simulation’s timeframes for the studied systems. We hereby infer that the mutation induced a more stable conformational structure on the neuraminidase protein.

### 3.2. Root Mean Square Fluctuation (RMSF)

The root mean square fluctuation (RMSF) is calculated to predict the changes in a protein ‘s conformations based on each residues contribution. In RMSF Equation (2) below, *x_i_*_(*j*)_ is the *i*-th Cα atom position in the *j*-th model structure, and (*x_i_*) represents the averaged location of the *i*-th Cα backbone atom in all models.
(2)$$RMSF=\sqrt{\frac{1}{n}\overset{n}{\underset{j}{\sum }}|x_{i\left(j\right)}-(x_{i}}){|}^{2}\;$$

In any given protein, each amino acid residue shares a common backbone which is made up of alpha carbon, carboxyl and amine moiety. These amino acids differ due to the different side chains that take varied sizes and atomic conformations. When a ligand molecule is in a complex with the protein, it restricts the movement of the side chains within the active site region. This differs from the root mean square deviation, averaged over time with specific values for each residue. The root mean square fluctuations were evaluated for Cα atoms of each amino acid of the two complexes (Figure 3a). We observed conformational flexibility with a similar trend in both the wildtype and the mutant complexes. However, the wildtype complex showed higher fluctuations from the beginning to the end of the simulations, while the mutant complex maintained a lower fluctuation during the entire simulation period. This suggests that the Val119 mutation induces higher stability to the peramivir–neuraminidase complex and further confirms the root mean square deviation results.

### 3.3. Radius of Gyration (RoG)

The radius of gyration is associated with compound stability and connected with the secondary structure of biomolecular systems [46]. It quantitatively measures the compactness, shape and folding of biomolecular compounds during molecular dynamic simulations. It is the moment of inertia of atoms from their centre of mass, which quantifies the molecular rigidity [47]. The radius of gyration is the square root of the inertia moment of atoms (Equation (3)), where *n* is the number of atoms, *r_i_* depicts the atomic position and *r_m_* represents the mean position of all atoms. Herein, we evaluated the RoG of peramivir wildtype and E119V mutant complexes, as depicted in Figure 4, using CPPTRAJ within the AMBER 18 suite [48].
(3)$$RoG=\sqrt{\frac{1}{n}\overset{i=0}{\underset{n}{\sum }}(r_{i}-r_{m}}{)}^{2}\;$$

The plot (3b) depicts that the wildtype exhibited significantly higher values from the beginning to the end of the 250 ns simulations. Conversely, the E119V mutant complex maintained lower values and consequently good compactness throughout the simulations. The substitution of the amino acid glutamate by valine in the E119V mutant resulted in more compactness and good folding in the peramivir neuraminidase complex.

### 3.4. Hydrogen Bond Analysis

Hydrogen bonds perform essential roles in molecular recognition and the overall stability of the protein structure. The formation of hydrogen bonds between amino acid residues is vitally essential in maintaining the structural conformation of proteins. When proteins undergo mutation, the hydrogen bonds are usually altered around the mutation site. The plot (Figure 4) depicts an apparent loss in hydrogen bond formation from 80 ns to 250 ns in the E119V mutant complex, thereby resulting in a distortion in the conformation of the wildtype complex.

To further investigate the impact of peramivir binding on the wildtype and E119V mutant systems, the evolution of hydrogen bond distances and the percentage occupancy of each intermolecular hydrogen bond were monitored between amino acid residues interacting with wildtype and E119V in the active site for 250-ns simulations (Table 1). The primary residues constituted hydrogen bonds Glu 119, Asp 151, Glu 213, Glu 278, Arg 371 and Tyr 406. Such amino acids were identified as key residues for neuraminidase–peramivir binding. The more vital interaction between peramivir and these residues plays an essential role in the peramivir’s potency.

The findings in Table 1 agree with the ligand–protein amino acid interaction at the active site in Figure 4, showing the atoms responsible for the hydrogen bond interactions. The mutant Glu 119–peramivir complex (87.36%) exhibited a high hydrogen bond percentage occupancy when compared to the wildtype Val 119–peramivir complex (79.26%) throughout the simulation. The mutant Glu 278–peramivir complex (88.83%) also indicated a high hydrogen bond percentage occupancy compared to the wildtype Glu 278–peramivir complex (52.40%). The E119V mutant Tyr 406–peramivir complex (97.13%) showed a high hydrogen bond percentage occupancy compared to the wildtype Tyr 406–peramivir complex (41.05%). These results suggest a strong interaction between E119V mutant active site residue atoms and peramivir compared to the wildtype. This could imply that interactions of E119V mutant amino acids with peramivir could be significant for high-affinity binding and relative stability of the peramivir–neuraminidase complex.

### 3.5. Principal Components Analysis (PCA)

The principal components analysis (PCA) makes the atomic dimensions of data from molecular dynamic simulation’s trajectories more concise, making it possible to visualise and analyse them while comparing the detected dynamic movements during the MD run. It diagonalises positions of the covariance matrix while identifying the orthogonal group of eigenvectors (modes) that describe the path of maximal variation in the detected conformational distributions. Therefore, slow alterations in conformations are detected when these prevalent modes are projected back to the original trajectory data [49,50]. Preceding the molecular dynamic simulation’s trajectory data analysis for PCA, solvents and ions were removed (stripped) from the PTRAJ Amber18 module, thereafter aligning the trajectories against a complete minimised structure. The principal component1 (PC1) and principal component2 (PC2) were analysed (Figure 5) while generating covariance matrices for the two systems (wildtype and E119V mutant) using origin software [46]. Figure 5 depicts a scattered plot for the two complexes, showing remarkable differences between the peramivir wildtype and E119V mutant complexes. The two systems displayed a conformational distribution in space. However, the E119V mutant complex exhibited slightly lower conformational changes than the wildtype.

### 3.6. MM/GBSA Binding Free Energy Calculation

The binding free energy analysis involves the endpoint energy calculations, which provide useful details about the ligand–protein complex association. In spontaneous processes, ligand–protein complexes occur when the change in Gibbs free energies (ΔG) of the given systems are negative and result when the systems reach equilibrium states at constant pressures and temperatures. Given that the ligand–protein association are subject to the magnitude of the negative ΔG, it is suggested that ΔG controls the stability of any given ligand–protein complex [44,46]. Therefore, the binding free energy is determined by the states of a system and, as thus, ΔG is determined by the initial and final thermodynamic states, irrespective of the pathway linking these states. The binding free energies of neuraminidase–peramivir wildtype and E119V mutant were computed using the molecular mechanics/generalized Born surface area method. The following equations, therefore, summarise the binding free energy:ΔG_bind_ = G_complex_ − G_receptor_ − G_ligand_(4)
ΔG_bind_ = E_gas_ + G_sol_ − TΔS(5)
E_gas_ = E_int_ + E_vdW_ + E_ele_(6)
G_sol_ = G_GB_ + G_SA_(7)
G_SA_ = γSASA(8)

### 3.7. MM/GBSA Binding Free Energy Calculation

Table 2 depicts the predicted MMGBSA binding energies for peramivir-wildtype (WT) and E119V mutant, respectively. The calculated total binding energy (ΔG_bind_) for peramivir–wildtype was −49.09 ± 0.13 kcal/mol, while the E119V mutant was −58.55 ± 0.15 kcal/mol. This difference is puzzling since atomistic modelling of these complexes in the gas phase (ΔG_gas_) show that WT is slightly energetically favoured than E119V. The decomposition of the binding energies into van der Waals, electrostatics, nonpolar and polar components enable us to identify the parameters driving the binding (Table 2) of the two complexes. We trace the solubility (ΔG_sol_) difference to the polar nature of the WT with glutamate over the hydrophobic valine of the mutant with ΔG_pol_ values, 151.11 ± 0.26 and 139.56 ± 0.24 kcal/mol, respectively. Additionally, the van der Waals binding energy contribution of −29.19 ± 0.08 kcal/mol slightly drives the higher calculated binding energy in the E119V mutant. Our calculation revealed the impact of explicit solvation on protein dynamics, energy prediction and overall remodification of the protein behaviour, mimicking an ideal system [51].

### 3.8. Per-Residue Contribution to Binding Free Energies

We further broke down the calculated binding energies for each active site residue to ascertain their significant intermolecular interacting contributions to the peramivir wildtype and E119V complexes. The active site residue total energies, electrostatic interactions, and the van der Waals energy contributions in kcal/mol are given in Figure 6A,C, respectively. Figure 6B,D also show the ligand interactions at the active site. In both systems, Arg 371 displayed the highest electrostatic contributions of −29.8 kcal/mol (wildtype) and −30.6 kcal/mol (mutant), respectively. There is a significant decrease in the electrostatic contribution in Glu 119 from −16.0 kcal/mol for the wildtype to −3.5 kcal/mol for the mutant. However, the electrostatic and van der Waal contributions from other residues in the mutant complex displayed significantly higher energies than in the wildtype. The overall higher van der Waal contribution and favourable higher hydrogen bond occupancy are suggested to be responsible for the higher compactness and stability in the mutant complex.

## 4. Conclusions

We investigated the intermolecular mechanism and dynamism of avian influenza virus H7N9 susceptibility to E119V-substituted peramivir–neuraminidase complex. The study employed the post-molecular dynamic simulation analysis in proffering multidimensional insights on the effects of the substituted 119V towards peramivir. The analysed total binding energy, PCA, RMSD, RMSF, RoG and hydrogen bond formation alludes to the fact that the E119V mutation conferred relative stability to the peramivir–neuraminidase complex. Our investigation shows that the effects of the substituted 119V amino acid residue increased the peramivir binding energy by −9.46 ± 0.02 kcal/mol. This increase in the total binding energy of the peramivir–neuraminidase wildtype from −49.09 ± 0.13 kcal/mol to −58.55 ± 0.15 kcal/mol, and the significantly increased hydrogen occupancy of the E119V, could be responsible for the consistent higher stability of the mutant complex, as shown in the different post analyses plots and diagrams. This study promises to proffer contributory insight and additional knowledge that would enhance future drug designs and help in the fight targeted at controlling the avian influenza H7N9 virus. Therefore, we suggest the need for an experimentalist to always collaborate with computational chemists, as we have carried out in our previous studies. This promises to save time and resources.

## Figures and Tables

**Figure 1 molecules-27-01640-f001:**
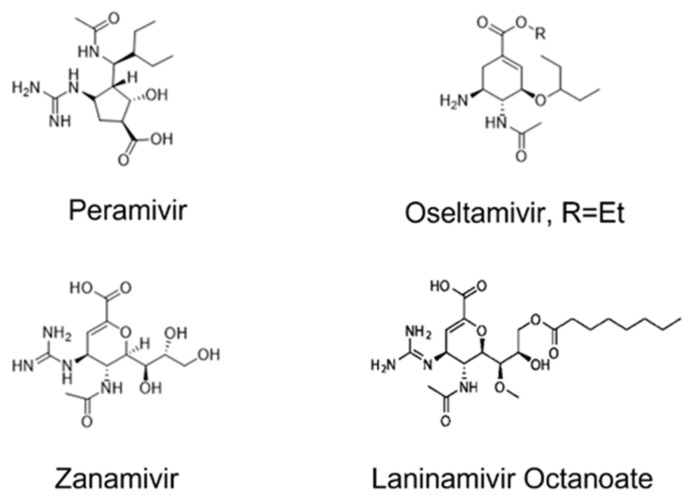
2D structures of neuraminidase inhibitors.

**Figure 2 molecules-27-01640-f002:**
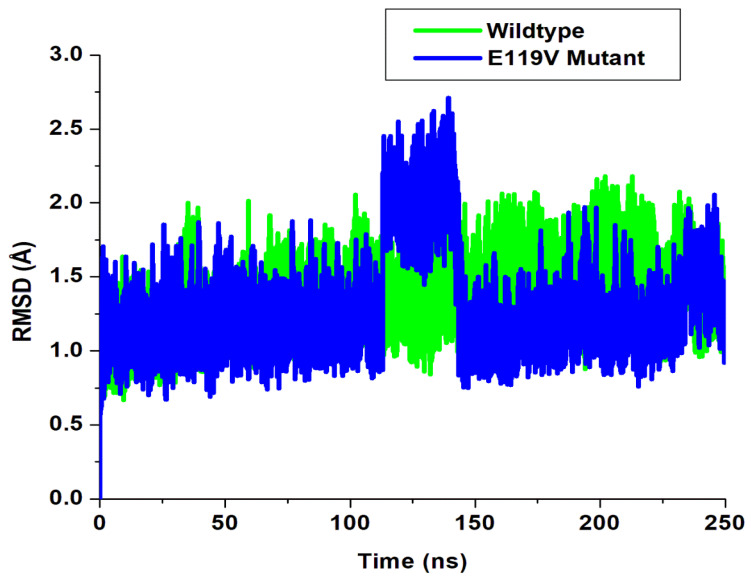
A plot of root mean square deviation.

**Figure 3 molecules-27-01640-f003:**
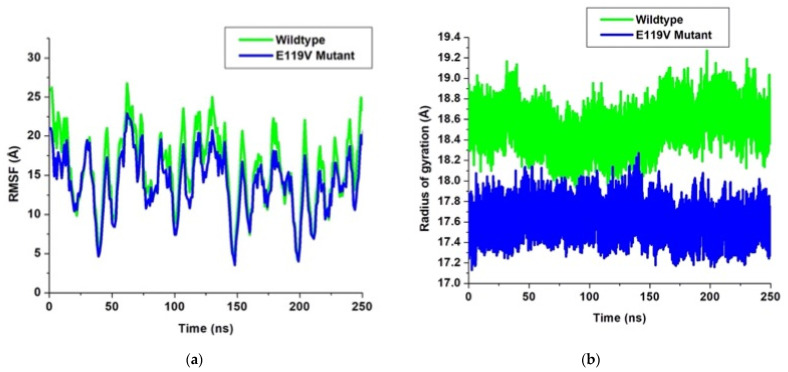
(**a**) Plots of root mean square fluctuation and (**b**) radius of gyration.

**Figure 4 molecules-27-01640-f004:**
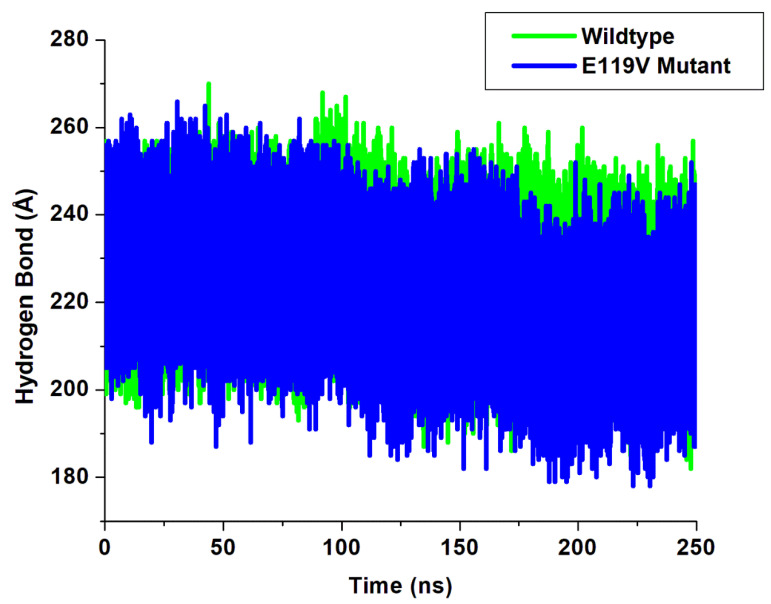
Plot of H-bond formation.

**Figure 5 molecules-27-01640-f005:**
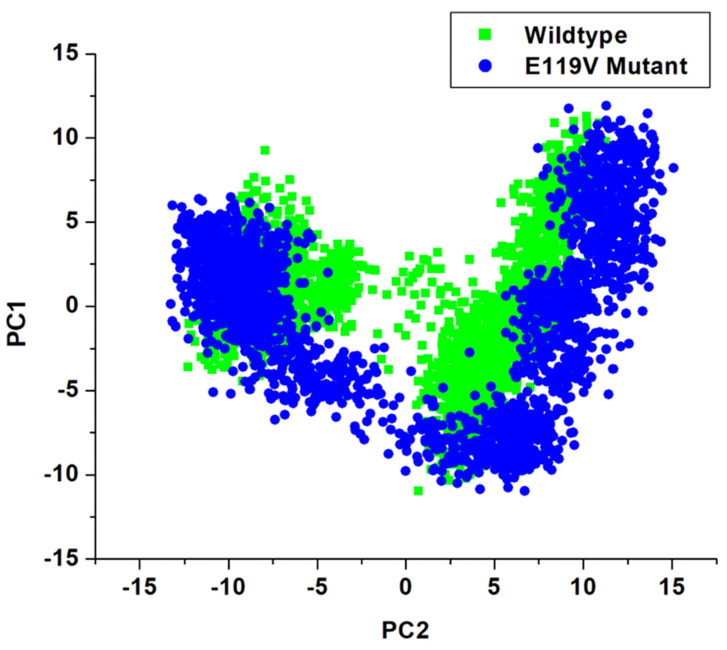
PCA scatter plot projection of C-α atoms motion of the first two principal components, PC1 and PC2.

**Figure 6 molecules-27-01640-f006:**
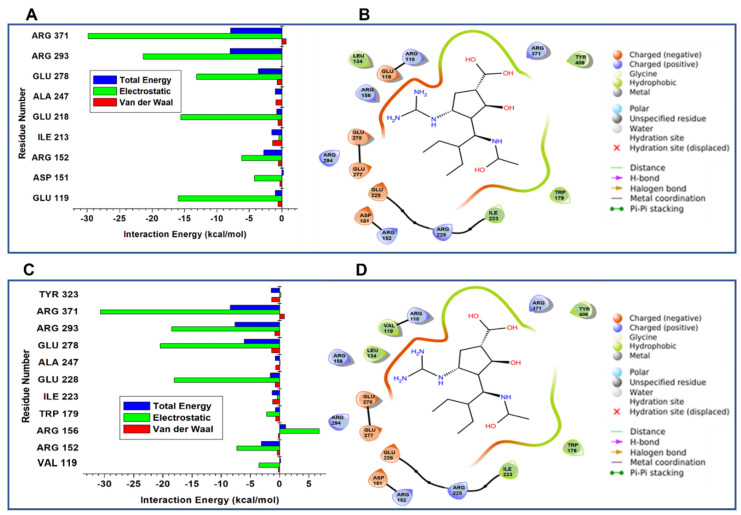
The electrostatic plots of van der Waals and the total energy per residue contributions for the wildtype (**A**) and the E119V mutant (**C**) complexes. The residues interacting at the active sites are also represented in (**B**,**D**).

**Table 1 molecules-27-01640-t001:** Percentage (%) occupancy and average distance (Å) between the peramivir (PERA) and prominent active site residues were calculated over simulation time.

Complexes	Acceptor	DonorH	Donor	Frames	Percentage Occupancy	Average Distance
	PERA@O14	Glu119@HH11	Glu119@NH1	123148	79.26	2.8135
Wildtype	PERA@O7	ARG371@HH22	ARG371@NH2	122459	68.21	2.8329
	GLU278@OE2	PERA@H25	PERA@N25	81010	52.40	2.8470
	TYR406@OH	PERA@H25	PERA@N25	27630	41.05	2.8491
	ASP151@OD1	PERA@H272	PERA@N27	6595	12.64	2.8949
	PERA@O8	ARG371@HH12	ARG371@NH1	147426	99.06	2.7693
E119V	TYR406@OH	PERA@H25	PERA@N25	142609	97.13	2.8687
	PERA@O14	VAL119@HH21	VAL119@NH2	117978	87.36	2.7971
	GLU278@OE2	PERA@H271	PERA@N27	121895	88.83	2.8190
	GLU213@OE2	PERA@H271	PERA@N27	55786	62.35	2.8874

**Table 2 molecules-27-01640-t002:** Calculated MMGBSA binding free energy for peramivir–wildtype and the E119V mutant.

Complexes	ΔG_vdv_	ΔE_ele_	ΔE_bind_	ΔE_gas_	ΔG_sol_	ΔG_pol_	ΔG_nonpol_
Wildtype	−26.86 ± 0.08	−168.21 ± 0.35	−49.09 ± 0.13	−195.07 ± 0.33	145.99 ± 0.26	151.11 ± 0.26	−5.12 ± 0.00
E119V	−29.19 ± 0.08	−163.77 ± 0.37	−58.55 ± 0.15	−192.96 ± 0.35	134.41 ± 0.24	139.56 ± 0.24	−5.15 ± 0.00

ΔG_bind_—binding free energy; ΔE_ele_—electrostatic interaction; ΔE_vdW_—van der Waals forces; ΔG_gas_—gas–phase interaction; ΔG_sol_—solvation energy; ΔG_pol_—polar salvation energy; ΔG_nonpol_—nonpolar salvation energy.

## Data Availability

The data presented in this study are available on request from the corresponding author.

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
