# Peer review of "Intermolecular Mechanism and Dynamic Investigation of Avian Influenza H7N9 Virus’ Susceptibility to E119V-Substituted Peramivir–Neuraminidase Complex"

_molecules, 2022, doi:10.3390/molecules27051640_

Round 1

Reviewer 1 Report

With the study entitled “Intermolecular Mechanism and Dynamic Investigation of Avian Influenza H7N9 Virus Susceptibility to E119V Substituted Peramivir-Neuraminidase Complex”, the authors explained  the mechanism and dynamism on the susceptibility of the E119V mutation on the peramivir-neuraminidase complex relative to the wild-type complex at the intermolecular level by using computerized algorithm, however, the current evidence does not fully elucidate phenomenon the mechanisms involved.

  1. “The human avian Influenza AH7N9 virus is linked with the seasonal flu” in abstract and introduction, which is not suitable. Because H7N9 virus is avian to human transmission but not human to human, while seasonal flu is typical human to human transmission. Moreover, H7N9 virus case is almost not reported after 2018, whereas seasonal flu outbreak every year.
  2. in abstract, the background should be succinct.
  3. The Introduction. should be succinct, Figure 1 an 2 is not needed.
  4. the experiment data in vitro about antiviral effect of peramivir on different H7N9 virus strains after E119V mutation should be added, which is key phenomenon before explain the mechanism by using computerized algorithm.
  5. The information of H7N9 strains is not clear.

Author Response

Point 1: “The human avian Influenza AH7N9 virus is linked with the seasonal flu” in abstract and introduction, which is not suitable. Because H7N9 virus is avian to human transmission but not human to human, while seasonal flu is typical human to human transmission. Moreover, H7N9 virus case is almost not reported after 2018, whereas seasonal flu outbreak every year.

Response 1: Thank you very much for pointing this out and also for providing a better insight, the sentence has been removed from the abstract (line 14) and the introduction of this manuscript ( 62-63, references 8 & 9).

Point 2: in abstract, the background should be succinct.

Response 2: The abstract background has been succinctly presented by the removal of lines 14-17.

Point 3: The Introduction. should be succinct, Figure 1 an 2 is not needed.

Response 3: The introduction has been made more succinct by removing figures 1 and 2. It is further reduced by the removal of lines 58-59, 71-74 and references 6, 8, 9, and 13.

Point 4: the experiment data in vitro about antiviral effect of peramivir on different H7N9 virus strains after E119V mutation should be added, which is key phenomenon before explain the mechanism by using computerized algorithm.

Response 4: Thank you very much for pointing out this. We acknowledged the importance of the experimental data and have always advocated the collaboration of experimentalist and computational chemist in other to save time, resources, and energy. This we also spell out in the conclusion part of this manuscript lines 355-357. However, E119V has been experimentally reported on neuraminidase-oseltamivir as could be seen “The single mutation E119V has clinically been reported in oseltamivir, an analogue of peramivir. However, the molecular dynamics and conformational alterations resulting from E119V substitution on peramivir is yet to be investigated. It is therefore imperative to further study the effects of this mutation on peramivir, an analogue of oseltamivir at the intermolecular level” (lines 116-120). Oseltamivir and Peramivir have been widely used in the treatment of H7N9 virus and given that these ligands have common receptor (neuraminidase) where the E119V mutation took place. We also considered the role computational biomolecular modelling played in the speedy insight into drug design against Covid-19, therefore, we delved into this study to provide insight into this E119V mutation on peramivir to guide our group on an ongoing experimental work leveraging on the intermolecular and interatomic effects of this mutation on peramivir to help save further experimental resources and time.

Point 5: The information of H7N9 strains is not clear.

Response 5: We truly appreciate your insight and your contribution in point 1, this has also helped in adjusting and correcting the information on the H7N9. Lines 55-56 and reference 7 are removed. In addition, lines 62-63 and references 8 and 9 are removed.

Reviewer 2 Report

In this manuscript, the mechanism and dynamism on the susceptibility of the E119V mutation on the peramivir-neuraminidase complex relative to the wild- type complex were investigated at the intermolecular level. The results were very interesting, which will provide an important reference for the development of new antiviral drugs. In my opinion, I recommend the manuscript for publication in Molecules after some modifications.

1、The effect two mutations on the interaction may be more interesting. The mechanism and dynamism on the susceptibility of the two mutations ( Arg292 to Lys292 and Glu 119 to Val119)  relative to the wild- type complex should also be investigated in detail.

Author Response

Point 1: The effect two mutations on the interaction may be more interesting. The mechanism and dynamism on the susceptibility of the two mutations ( Arg292 to Lys292 and Glu 119 to Val119)  relative to the wild- type complex should also be investigated in detail.

Response 1: Thank you very much for thinking in the same line of thought with our group. We have a study on Arg292 to Lys292 on peramivir already accepted for publication in Molecule-MDPI journal and another ongoing study on the suggested double mutation.

Reviewer 3 Report

This paper is meaningful as a study that verified the effectiveness of treatments according to the mutation of nueraminidase of influenza virus H7N9 through computer modeling.
I think it will be very helpful in the development of virus treatments.
However, as the author mentioned, some additional content seems to be needed.
1. If there is a paper that has experimentally confirmed the effectiveness of the treatment through a combination of therapeutic agent, enzyme, or protein structure using the program analyzed in this study, the reliability of the study should be increased by adding it as a reference.
2. In the case of peramivir, it is used as an injection through intravenous injection, so it should be mentioned about the environment of blood because it requires structural binding with viral proteins in the blood.

Author Response

Point 1: If there is a paper that has experimentally confirmed the effectiveness of the treatment through a combination of therapeutic agent, enzyme, or protein structure using the program analyzed in this study, the reliability of the study should be increased by adding it as a reference.

Response 1: Thank you very much for raising this important point. “Donald et al. (2010) carried out a combinatory research on peramivir and oseltamivir. The conducted experimental study on the chemotherapeutic effects of the combined treatment on infected with the influenza virus. The treatment was carried out with peramivir and oseltamivir for three days. The result showed that such combination yielded an improved outcome better than treatment with suboptimal doses of peramivir and oseltamivir in the treatment of influenza virus in mice”. This has been added in the manuscript (lines 80-85) and reference 16. We shall also consider carrying out another separate computational investigation on this combination therapy. We sincerely appreciate your point.

Point 2: In the case of peramivir, it is used as an injection through intravenous injection, so it should be mentioned about the environment of blood because it requires structural binding with viral proteins in the blood.

Response 2: Thank you again for providing a good insight that is useful in strengthening this manuscript, we have added in lines 92-105 and references 17-23. ” The structure of peramivir consists of carboxylic and guanidino moieties in addition to the lipophilic side chain of the cyclopentane backbone. These structural moieties enable peramivir to establish stronger bond with the N9 of the neuraminidase enzyme with low dissociation rate. Zhang et al. investigated the pharmacokinetic properties of peramivir in selected healthy Chinese volunteers. Three hundred to four hundred milligram doses of peramivir were intravenously administered on them. When blood samples were collected at intervals after administering the doses, it was discovered that peramivir exhibited linear pharmacokinetics with an observed increase in maximum concentration. In another study by Sato et al., aimed at determining and predicting peramivir concentration-time curve of the H7N9 virus reduced susceptibility to neuraminidase inhibitors. Ten milligram per kilogram of peramivir was intravenously administered to paediatric patients, peramivir resulted in reduced maximal concentration of about 0.1 percent within 24 hours of the drug administration. The study suggested re-administering of peramivir in cases of delayed improvement in patients with normal susceptibility to influenza A and B and that better viral inhibition and lower frequency of adverse effects may be expected with divided administration”.

Round 2

Reviewer 1 Report

The current evidence does not fully support the conclution, the following questions should be adressed.  

  1. Although E119Vhas been experimentally reported on neuraminidase-oseltamivir, an analogue of peramivir, the related biology data don’t reflect the role of E119V on neuraminidase-peramivir. Therefore, the experiment data about antiviral effect of peramivir on H7N9 virus strains after E119V mutation should be illustrated, which is key  phenomenon.
  2. Which H7N9 strain that you used as your model should be added.

Author Response

Point 1: Although E119Vhas been experimentally reported on neuraminidase-oseltamivir, an analogue of peramivir, the related biology data don’t reflect the role of E119V on neuraminidase-peramivir. Therefore, the experiment data about antiviral effect of peramivir on H7N9 virus strains after E119V mutation should be illustrated, which is key  phenomenon.

Response 1: Thank you for this good point. We explained in the original submitted manuscript and in the revised version usefulness of experimental chemists collaborating with computational chemists. Researchers could choose either to start working first from experimental to computational validation and vice versa. This indeed has helped in saving the time that would have required for drug design process. Recent developments in the filed of drug design have witnessed this kind collaboration especially in the recent fight against COVID-19. We appreciate the raised concern and have also mentioned that  the authors are already working on the experimental aspect of this E119V mutations based on the results and the insights from this computational modelling.

See response to the revised version 

"Thank you very much for pointing out this. We acknowledged the importance of the experimental data and have always advocated the collaboration of experimentalist and computational chemist in other to save time, resources, and energy. This we also spell out in the conclusion part of this manuscript lines 355-357. However, E119V has been experimentally reported on neuraminidase-oseltamivir as could be seen “The single mutation E119V has clinically been reported in oseltamivir, an analogue of peramivir. However, the molecular dynamics and conformational alterations resulting from E119V substitution on peramivir is yet to be investigated. It is therefore imperative to further study the effects of this mutation on peramivir, an analogue of oseltamivir at the intermolecular level” (lines 116-120). Oseltamivir and Peramivir have been widely used in the treatment of H7N9 virus and given that these ligands have common receptor (neuraminidase) where the E119V mutation took place. We also considered the role computational biomolecular modelling played in the speedy insight into drug design against Covid-19, therefore, we delved into this study to provide insight into this E119V mutation on peramivir to guide our group on an ongoing experimental work leveraging on the intermolecular and interatomic effects of this mutation on peramivir to help save further experimental resources and time".

Point 2: Which H7N9 strain that you used as your model should be added

Response 2: A/Anhui/1/2013 H7N9

https://doi.org/10.1093/infdis/jit554

Reviewer 2 Report

The author has revised the article according to the opinions of the three reviewers, and the quality of the article has been greatly improved. I recommend the manuscript for publication in Molecules.

Author Response

thank you